# The influence of the antithymocyte globulin dose on clinical outcomes of patients undergoing kidney retransplantation

**Kamilla Linhares[1], Julia Bernardi Taddeo[1], Marina Pontello Cristelli[1], Henrique Proença[1], Klaus Nunes Ficher[1], Renato de Marco[2], Maria Gerbase-DeLima**  **[2], Jose Medina-Pestana[1], Helio Tedesco-Silva**  **[1] \***

**1** Division of Nephrology, Hospital do Rim, Universidade Federal de São Paulo, São Paulo, Brazil,
**2** Immunogenetics Institute, Associação Fundo de Incentivo à Pesquisa, São Paulo, Brazil

\* heliotedesco@medfarm.com.br

**Data Availability Statement:** All relevant data are within the paper and its supporting information files.

## Abstract

Optimizing antithymocyte globulin (rATG) dosage is critical for high immunological risk patients undergoing a repeat kidney transplant. This natural retrospective cohort study compared clinical outcomes of two successive cohorts of consecutive recipients of retransplants receiving 5 x 1 mg/kg (rATG-5, n = 100) or a single 3 mg/kg (rATG-3, n = 110) dose of rATG induction therapy. All patients had negative complement-dependent cytotoxicity crossmatch and no anti-HLA A, B, DR donor-specific antibodies (DSA). The primary endpoint was efficacy failure (first biopsy-proven acute rejection, graft loss, or death) at 12 months. There was no difference in the cumulative incidence of efficacy failure (18.0% vs. 21.8%, HR = 1.22, 95% CI 0.66–2.25), respectively. There were no differences in 3-years freedom from biopsy proven acute rejection, and patient, graft, and death-censored graft survivals. There were no differences in the incidence of surgical complications (25.0% vs. 18.2%; p 0.151), early hospital readmission (27.8% vs. 29.5%; p = 0.877) and CMV infections (49% vs. 40%; p = 0.190). There were also no differences in the incidence (59.6% vs. 58.7%, p = 0.897) and duration of delayed graft function but a stable difference in estimate glomerular filtration rate was observed from month 1 (54.7±28.8 vs. 44.1±25.3 ml/min/1.73 m$^2$, p = 0.005) to month 36 (51.1±27.7 vs. 42.5±24.5, p = 0.019). Mean urinary protein concentration (month 36: 0.38±0.81 vs. 0.70±2.40 g/ml, p = 0.008) and mean chronic glomerular Banff score in for cause biopsies (months 4–36: 0.0±0.0 vs. 0.04±0.26, p = 0.044) were higher in the rATG-3 group. This cohort analysis did not detect differences in the incidence of efficacy failure and in safety outcomes at 12 months among recipients of kidney retransplants without A, B, and DR DSA, receiving induction therapy with a single 3 mg/kg rATG dose or the traditional 5 mg/kg rATG.

## Introduction

The number of patients requiring repeat kidney transplants is increasing [1, 2]. According to the most recent OPTN/SRTR 2019 Annual Data Report, 16.3% (2009), 14.4% (2014), and

**Funding:** KL has received a research grant from Conselho Nacional de Desenvolvimento Científico e Tecnológico (CNPq) - Brasil, to carry out the present work. The funders had no role in study design, data collection and analysis, decision to publish, or preparation of the manuscript.

**Competing interests:** I have read the journal's policy and the authors of this manuscript have the following competing interests: Helio Tedesco-Silva has received speaker's fees and travel or accommodation expenses for development of educational presentations and scientific advice from Novartis, Pfizer, and Roche. Jose Medina Pestana has received speaker's fees and travel or accommodation expenses for development of educational presentations and scientific advice from Bristol-Myers Squibb, Novartis, Pfizer, and Roche. Marina Pontello Cristelli has received speaker's fees for development of educational presentations and travel or accommodation expenses from Novartis and Pfizer. The other authors of this manuscript have no conflicts of interest to disclose. This does not alter our adherence to PLOS ONE policies on sharing data and materials.

**Abbreviations:** BPAR, Biopsy proven acute rejection; CI, Confidence Interval; CKD-EPI, Chronic Kidney Disease Epidemiology Collaboration Equation; CMV, Cytomegalovirus; CsA, Cyclosporine; DGF, Delayed graft function; DSA, Donor specific antibody; eGFR, Estimated glomerular filtration rate; MFI, Mean fluorescence intensity; HBsAg, Hepatitis B Surface Antigen; HCV, Hepatitis C Virus; HIV, Human Immunodeficiency Virus; HLA, Human Leukocyte Antigen; OPTN/SRTR, Organ Procurement and Transplant Network/Scientific Registry of Transplant Recipients; PRA, Panel Reactive Antibody; rATG, rabbit antithymocyte globulin; USRDS, United States Renal Data System.

11.8% (2019) of the patients on the waiting list had a history of a previous transplant [3]. Traditionally, these patients are considered to have a high immunological risk for early acute rejection and graft loss [4]. Particular to this specific population is the previous sensitization to HLA antigens and possible reexposure to mismatched HLA antigens, even with a negative crossmatch [5]. Consequently, induction therapy with lymphocyte depleting agents is recommended [6]. Among several protocols, the use of 5 consecutive 1 mg/kg daily doses of rATG 5 mg/kg, adjusted based on either total or CD3 positive lymphocyte counts, is routinely used [7]. Specifically among patients undergoing kidney retransplants, a recent analysis of data from 14,336 patients extracted from the United States Renal Data System (USRDS) registry showed no significant differences between induction groups for outcomes of delayed graft function, 1-year acute rejection, 1-year BK virus or patient death [8]. More recently, reduced rATG dosing regimens have been proposed based on immunological risk stratification. Cumulative rATG doses of 3 mg/kg were administered to nonsensitized living donor recipients, 4.5 mg/kg to nonsensitized deceased donor recipients and 6 mg/kg to higher immunologic risk recipients, including those with history of prior transplant. One-year rejection rates in the first 2 groups were 8.3% and 8.8%, respectively [9].

The assessment of immunological risk has been improving with the development of new tools and better mechanistic understanding of immune reactions in sensitized patients. The routine use of solid phase assays provides detailed analysis of the presence of preformed anti-HLA antibodies, allowing for the implementation of allocation policies with reduced immunological risk [10].

On Aug 1st, 2009, changes in the allocation system for recipients of deceased donor kidney allografts were implemented in our center. Besides the negative complement-dependent cytotoxicity (CDC) crossmatch, all patients undergoing kidney transplantation should not have pre-formed anti-HLA A, B and DR donor specific antibodies with mean fluorescence intensity (MFI) above 1500. Inherently, this strategy is associated with a lower risk for early acute rejection and graft loss [11]. Based on this assumption, we decided to change our induction protocol by reducing the dose of rATG from 5 to a single 3 mg/kg dose since Jun 16th, 2014, after determination of its safety and efficacy in a randomized trial [12]. Our hypothesis was that in high immunological risk patients undergoing retransplantation without pre-formed HLA A, B, DR donor specific antibodies, the use of a single 3 mg/kg dose of rATG would provide comparable efficacy to the traditional 5 mg/kg dose, with possible safety benefits related to surgical complications, number of early hospital readmissions (EHR) and incidence of CMV infection.

## Materials and methods

### Study design

This was a sequential cohort experiment in which all kidney transplant recipients started receiving 3 mg/kg of rATG for induction therapy from June 17th, 2014, at our institution. Therefore, two retrospective cohorts of consecutive adult patients undergoing repeat kidney transplants were constructed. We identified all consecutive kidney retransplants who received 5 mg/kg r-ATG (rATG-5 group) from January 1st, 2010 to June 16th, 2014 and all consecutive kidney retransplants who received 3 mg/kg r-ATG (rATG-3 group) from June 17th, 2014 to October 9th, 2016. Data was collected up to October 31st, 2019. There were no further changes in the protocols or in clinical practice over the time frame of this study. We compared the incidence of acute rejection, patient and graft survivals, and the incidence of early hospital readmissions, surgical complications and CMV infection.

## Ethics

Kidney transplants were performed at the Hospital do Rim with organs recovered from deceased donors under the National and São Paulo State secretary allocation policy, and from living donors, according to the National legislation. The national regulation does not mandate registration of donors. None of the transplant donors was from a vulnerable population and all donors or next of kin provided written informed consent that was freely given. Informed consent was obtained from all donors or their next of kin, prior to organ recovery. Informed consent forms for deceased (S1 Appendix: informed consent form for solid organs and tissues donation–deceased donor–over 18 years-old) and living donors (S2 Appendix: Term of authorization for kidney donation from transplantation) are provided. According to Brazilian legislation, no medical costs or other cash payments are allowed to the family of the donor. Record data from kidney transplants performed between 01Jan2010 to 09Oct2016 were screened for this analysis. The anonymized data set necessary to replicate our study findings is provided (S3 Appendix: Master File). The research project was approved by the local Research Ethics Committee of the Federal University of São Paulo (CAAE 02285018.2.5505). The IRB waived the requirement to obtain informed consent because of the retrospective nature of the record review, the lack of interference with the rights and welfare, and the lack of risk to the participants. All data were retrieved from the electronic medical records and were fully anonymized before analysis. Medical records of the Hospital do Rim (Fundação Oswaldo Ramos) were accessed between 01/06/2017 and 01/10/2020.

## Study population

All patients in both cohorts had a negative complement dependent cytotoxicity crossmatch, no preformed A, B and DR donor specific antibodies with mean fluorescence intensity higher than 1500 in both historical and current sera and had received an ABO compatible repeat renal allograft from living or deceased donors. To identify anti-HLA antibodies in serum, flowPRA screening test was performed, followed by single antigen bead assay to identify the specificity of anti-HLA antibodies, using the Luminex platform. Deceased donor kidneys were allocated based on HLA compatibility, with a single priority criterion related to the imminent technical impossibility to obtain access to perform any type of dialysis. We excluded recipients of simultaneous kidney and pancreas transplants, pediatric kidney transplant recipients, patients who received basiliximab induction and patients who did not receive induction or with no available data on induction and those who received CNI-free immunosuppressive regimens.

## Immunosuppression protocols

**Induction therapy.**   During the first period, all patients with PRA > 50% or with other risk factors such as low HLA compatibility (more than 3 HLA mismatches), long cold ischemia time (higher than 24 hours), priority criterion (priority status attributed to a patient with imminent lack of access for peritoneal hemodialysis during the allocation process), and/or donor type (expanded donor criteria) received the standard of care induction therapy consisting of five 1 mg/kg doses of rATG. The initial 1.0 mg/kg dose of rATG, administered intravenously over 8 hours, beginning within the first 24 hours after graft revascularization. The subsequent 4 daily doses were adjusted based on the peripheral lymphocyte counts ($<$ 100 cells/mm$^3$: hold the dose; 100–150 cells/mm$^3$: reduce 25 mg; 150–300 cells/mm$^3$: maintain the dose; $>$300 cells/mm$^3$: increase 25 mg). During the second period, all patients, regardless of the value of PRA, received a single 3mg/kg dose of rATG, administered intravenously over 10 hours, beginning within the first 24 hours after graft revascularization. In both periods, the rATG dose was based on current dry weight, rounded to the 25 mg vial size.

**Maintenance therapy.** Patients received tacrolimus (TAC) 0.1 mg/kg twice daily starting on day 1, with doses adjusted to maintain whole blood trough concentrations between 5–15 ng/ml, in combination with mycophenolate sodium 720 mg twice daily or azathioprine 2 mg/kg daily. Patients received TAC 0.05 mg/kg twice daily, with doses adjusted to maintain blood concentrations between 3–5 ng/ml, in combination with everolimus 1.5 mg twice daily, to maintain whole blood trough concentrations between 3 to 8 ng/ml. Patients received cyclosporine (CsA) 5 mg/kg twice daily, with doses adjusted to maintain whole blood trough concentrations between 100 to 300 ng/ml in combination with mycophenolate sodium 720 mg twice daily or azathioprine 2 mg/kg daily. All recipients were initiated on intravenous methylprednisolone (500 mg) intraoperative, followed by 0.5 mg/kg oral PRED, tapered to daily 5 mg doses by the end of the first month. Maintenance drug combination was selected based on perceived risk assessment, considering the previous transplant outcomes and the immunological risk of the retransplant.

## Prophylaxis

All patients received a 5-day course of 400 mg of albendazole and continuous use of sulfamethoxazole-trimethoprim. None of the patients received pharmacological prophylaxis for CMV infection. Preemptive strategy was used for donor (+) and recipient (-) CMV serostatus combination (D+/R-), for patients receiving mycophenolate sodium and after treatment of acute rejection AR episodes. The other patients were monitored at the physician discretion. The preemptive therapy consisted of every other week monitoring the viral replication, from the third week after transplantation, until the end of the third month, using the CMV pp65 antigenemia assay [13]. During the study period, the CMV antigenemia test was performed in a single laboratory using the CMV Brite Turbo kit, according to the manufacturer's recommendations (IQ® Products, Groningen, Netherlands).

## Acute rejection

Treated acute rejection (tAR) included clinical acute rejections and biopsy proven acute rejection (BPAR). Clinical acute rejections were defined as graft dysfunction, without histological evidence of rejection and treated with methylprednisolone for at least three days. All episodes BPAR were adjudicated and retrospectively reclassified based on the Banff 2019 criteria. The cumulative survival-free of BPAR ($\geq$IA) were compared over 3 years of follow up.

## Graft loss and mortality

The cumulative incidence of graft loss, death-censored graft loss and mortality were compared over 3 years of follow up. Graft loss was defined as the need for permanent return to dialysis. Loss to follow-up due to referral to another transplant center was defined by the lack of information for more than 6 months.

## Safety outcomes

We selected three safety surrogate outcomes to compare the differential effect of the two rATG doses, namely, complications associated with the surgical procedure, early hospital admission for any cause and CMV events.

## Surgical complications

We evaluated the incidence, timing and recurrence of surgical complications between the two groups.

### Early hospital readmission

The incidence and specific causes of EHR, defined as all readmissions within 30 days of initial hospital discharge, were compared between the two groups.

### CMV infection or disease

CMV infection was defined as the presence of more than 10 infected cells in a total of 200,000 peripheral blood neutrophils in asymptomatic patients, based on CMV pp65 antigenemia assay. CMV disease was diagnosed based on the presence of CMV-related signs or symptoms including fever, asthenia, myalgia, leukopenia, thrombocytopenia, or liver enzymes abnormalities, and the presence of any number of CMV pp65 infected cells. CMV infection or disease was treated with intravenous ganciclovir for at least 14 days with weekly monitoring of viral replication. Treatment was continued for 1 week, after the first negative CMV pp65 antigenemia test.

### DGF and renal function and histology

Delayed graft function was defined as the need of dialysis during the first week after transplantation, excluding a single dialysis for hypervolemia and/or hyperkalemia. Duration of DGF was measured from the time of transplant to the last dialysis. Renal function was assessed by the estimated glomerular filtration rate (eGFR, mL/min/1.73 m$^2$) calculated with the Chronic Kidney Disease Epidemiology (CKD-EPI) study equation. Surveillance graft biopsies were performed during the DGF period and in patients with incomplete renal function recovery, as well as in all episodes of graft dysfunction.

### Primary endpoint

The primary endpoint was to compare the incidence of efficacy failure, defined by first biopsy proven acute rejection ($\geq$ IA), graft loss, or death between the groups, at 12 months.

### Secondary endpoints

Secondary endpoints included the comparison of the cumulative incidence of first biopsy proven acute rejection ($\geq$ IA), graft loss, death-censored graft loss and mortality over 36 months. We also analyzed the incidence and severity of treated acute rejection episodes, the incidence of surgical complications, early hospital readmission and CMV events during the first 12 months. Finally, we compared the incidence and duration of DGF, the trajectories of renal function, and proteinuria up to 36 months of follow up.

### Statistical analysis

Categorical variables were expressed as absolute frequency and percentages and the differences between groups were performed using the chi-square test or Fisher's exact test. Continuous variables were presented as mean and standard deviation or medians (interquartile range [IQR]), depending on normality. Differences between groups were identified using the Mann-Whitney test or Student's t-test. Renal function (eGFR) trajectories were compared up to 36 months using a generalized model for repeated measures, without and with imputation of a value of "zero" in cases of graft loss, and with the last observation carried forward analysis (LOCF) for patients who died or were loss to follow up. Multiple linear regression was used to identify independent risk factors associated with 1-month eGFR.

Considering the non-inferiority margin of 15% for the experimental rATG-3 group compared to the standard rATG-5 group for the primary endpoint of efficacy failure at 12 months, a significance level of 5%, a statistical power of 80%, and the efficacy failure of 25% in the

control arm, 208 patients are required to be 80% sure that the upper limit of a one-sided 95% confidence interval. We used Cox proportional hazard model to compare hazard functions using treatment group as factor. Non-inferiority was determined if upper confidence limit of HR was less than the non-inferiority margin then non-inferiority using Wald's confidence interval. The survival curves were obtained by the Kaplan-Meier method and the differences identified by the Log-Rank test, censored for patients with loss to follow-up. All comparisons were made using the intention to treat population, defined as patients receiving at least of dose of rATG in both groups.

All statistical analyzes were performed using the SPSS v.21 program (SPSS inc., Chicago, IL, USA), with two-tailed hypothesis testing and a α of 0.05 as a criterion for statistical significance.

## Results

### Study population

There were 4030 kidney transplants between January 1, 2010 and June 16, 2014, of which 203 (5%) were retransplants, while among 2098 procedures between June 17, 2014 and October 9, 2016, 139 (6.6%) were retransplants. We excluded 103 retransplants in the first period and 29 in the second period who did not receive induction therapy with rATG (S1 Fig). Therefore, we identified 100 consecutive retransplants who received 5 mg/kg and 110 receiving 3 mg/kg rATG. Patients were relatively young despite long time on dialysis and 13% had priority criterion. About 44% had received the first kidney transplant from a deceased donor and a similar proportion had undergone previous graft nephrectomy. The majority was sensitized and receiving a second transplant from a deceased donor. The proportion of patients with PRA class I or class II >50% was 65% (n = 65) in the rATG-5 group and 56% (n = 62) in the rATG-3 group. There were no differences in median cPRA (50% vs. 44%) but patients in the rATG-3 had lower median PRA class I. Importantly, 73% received a zero HLA DR mismatch and 14% a zero HLA A, B, DR mismatch kidney retransplants. Median KPDI was 60% and 19% were extended criteria donors. There was an imbalance in the distribution of donor/recipient mismatches favoring the rATG-5 group. The degree of acute kidney injury, as measured by the difference between the final and initial creatinine (Δ), was higher in the rATG-3 group. Finally, kidneys were transplanted with a median cold ischemia time between 22 and 24 hours (Table 1).

### Immunosuppression

As per protocol, there was a significant difference in the median dose of rATG (5.3 vs. 3.01 mg/kg, p<0.0001). In the rATG-5 group only 4 patients received less than 3 mg/kg, 32 received between 3 and 5 mg/kg, 32 received between 5 and 6 mg/kg, and 32 more than 6 mg/kg (S1 Table). The majority of the patients received tacrolimus in combination with mycophenolate sodium in both groups. Mean tacrolimus whole blood trough concentrations were lower in the rATG-5 at 1 month (10.2±3.6 vs. 11.6±4.5 ng/ml, p = 0.018) and 12 months (7.8±2.8 vs. 9.1 ±3.5 ng/ml, p = 0.008) while mean prednisone doses were higher at 1 month (17.4±6.4 vs. 12.4 ±5.4 mg/day, p<0.0001), 3 months (8.2±5.8 vs. 5.2±1.4 mg/day, p<0.0001) and 6 months (6.1 ±3.6 vs. 5.2±0.9 mg/day, p = 0.039) compared to rATG-3. There were no differences in mean mycophenolate sodium doses during the first 12 months after transplantation (Table 2).

### Efficacy failure

There was no significant difference in the incidence of efficacy failure at 12 months (18.0% in the rATG-5 vs. 21.8% in the rATG-3 group, HR = 1.22, 95% CI 0.66–2.25, Table 3, Fig 1). There was no difference in the incidence, severity, timing, and type of treatment or recurrent

**Table 1. Demographic characteristics of the study population.**

| Parameters | rATG-5 (n = 100) | rATG-3 (n = 110) | p value |
|---|---|---|---|
| **Recipient** | | | |
| Age (years), median (IQR) | 40 (32.5; 47) | 40 (30; 50) | 0.873 |
| Sex (male), n (%) | 54 (54) | 68 (61.8) | 0.251 |
| Donor/recipient | | | 0.043 |
| Donor male/recipient male | 35 (35) | 37 (33.5) | |
| Donor female/recipient male | 19 (19) | 31 (28) | |
| Donor male/recipient female | 30 (30) | 17 (15.5) | |
| Donor female/recipient female | 16 (16) | 25 (23) | |
| Race, n (%) | | | 0.657 |
| White | 51 (51) | 57 (51.8) | |
| Black | 16 (16) | 13 (11.8) | |
| Other | 33 (33) | 40 (36.4) | |
| CKD etiology, n (%) | | | 0.541 |
| Hypertension | 11 (11) | 10 (9.1) | |
| Glomerulonephritis | 34 (34) | 42 (38.2) | |
| Diabetes Mellitus | 3 (3) | 2 (1.8) | |
| In determined | 36 (36) | 42 (38.2) | |
| Urologic | 6 (6) | 9 (8.2) | |
| Polycystic kidney disease | 1 (1) | 2 (1.8) | |
| Other | 9 (9) | 3 (2.7) | |
| Renal replacement therapy, n (%) | | | 0.491 |
| Hemodialysis | 83 (83) | 83 (7.,5) | |
| Peritoneal dialysis | 2 (2) | 3 (2.7) | |
| Hemodialysis + Peritoneal dialysis | 15 (15) | 23 (20.9) | |
| Conservative | 0 | 1 (0.9) | |
| Viral serology, n (%) | | | 0.278 |
| HIV | 1 (1) | 1 (0.9) | |
| HBV | 0 | 4 (3.6) | |
| HCV | 9 (9) | 8 (7.3) | |
| CMV (donor/recipient) | | | 0.213 |
| Donor (+)/recipient (-) | 2 (2) | 0 (0) | |
| Donor unknown/recipient (-) | 1 (1) | 1 (0.9) | |
| Donor (+/-)/recipient (+) | 95 (95) | 109 (99.1) | |
| Donor unknown/recipient unknown | 2 (2) | 0 (0) | |
| Time on dialysis (months), median (IQR) | 71 (28; 118) | 53 (26; 121) | 0.238 |
| Priority criterion, n (%) | 16 (16) | 12 (10.9) | 0.278 |
| Previous transplant (deceased), n (%) | 44.9 (44) | 44 (40) | 0.475 |
| Previous graft nephrectomy | 46 (46) | 50 (45.5) | 0.937 |
| Current kidney transplant, n (%) | | | 0.356 |
| Second | 95 (95) | 101 (91.8) | |
| Third | 5 (5) | 9 (8.2) | |
| cPRA Class I, median (IQR) | 58.5 (19; 98) | 31.5 (0; 76) | 0.045 |
| cPRA Class II, median (IQR) | 23 (0; 57.5) | 18 (0; 53) | 0.509 |
| cPRA, median (IQR) | 50 (21.75; 78.25) | 44 (4.4; 83.6) | 0.111 |
| HLA mm, median (IQR) | 2 (1;3) | 2 (1;3) | 0.803 |
| zero HLA A mm, n (%) | 29 (29) | 29 (26.4) | 0.392 |
| zero HLA B mm, n (%) | 28 (28) | 29 (26.4) | 0.455 |

*(Continued)*

**Table 1.** (Continued)

| Parameters | rATG-5 (n = 100) | rATG-3 (n = 110) | p value |
|---|---|---|---|
| zero HLA DR mm, n (%) | 75 (75) | 78 (70.9) | 0.305 |
| zero HLA A, B, DR, n (%) | 15 (15) | 14 (12.7) | 0.390 |
| **Donor** | | | |
| Age (years), median (IQR) | 42 (31; 53) | 45.5 (37; 54) | 0.421 |
| Sex (male), n (%) | 65 (65) | 54 (49.1) | 0.020 |
| Race, n (%) | | | 0.826 |
| *White* | 60 (60) | 65 (59.1) | |
| *Black* | 14 (14) | 13 (11.8) | |
| *Other* | 26 (26) | 32 (29.1) | |
| Deceased donor, n (%) | 94 (94) | 99 (90) | 0.289 |
| *ECD criteria, n (%)* | 22 (22) | 17 (15.4) | 0.281 |
| Cause of death, n (%) | | | 0.767 |
| *Cerebrovascular* | 56 (59.6) | 60 (60,6) | |
| *Trauma* | 31 (33) | 31 (31,3) | |
| *Tumor SNC* | 2 (2.1) | 1 (1) | |
| *Anoxia* | 3 (3.2) | 2 (2) | |
| *Other* | 2 (2.1) | 5 (5.1) | |
| Previous cardiac arrest, n (%) | 18 (19.1) | 14 (14.1) | 0.350 |
| Use of vasoactive drug, n (%) | 82 (87.2) | 88 (88,9) | 0.723 |
| Infection, n (%) | 37 (39.4) | 46 (46.5) | 0.319 |
| Diabetes, n (%) | 5 (5.3) | 4 (4) | 0.674 |
| Hypertension, n (%) | 31 (33) | 38 (38.4) | 0.434 |
| Initial creatinine, mg/dl, median (IQR) | 1.0 (0.8; 1,2) | 0.8 (0.59; 1,01) | 0.070 |
| Final creatinine, mg/dl, median (IQR) | 1.28 (0.74; 1,82) | 1.4 (0.6; 2,2) | 0.134 |
| Δ creatinine, mg/dl, median (IQR) | 0.2 (-0.10; 0.85) | 0.5 (0; 1.4) | 0.019 |
| KDPI %, median (IQR) | 60 (36; 84) | 63 (46; 80) | 0.245 |
| Cold ischemia time (hours) median (IQR) | 22 (19; 25) | 24 (19,5; 28,5) | 0.060 |

Initial creatinine value is the first value obtained at the time of hospital admission of the potential donor. Final creatinine is the last creatinine value before organ recovery. Delta creatinine is an arbitrary measure of acute kidney injury. Δ creatinine = final creatinine–initial creatinine (75 paired samples in each group).

r-ATG: rabbit antithymocyte globulin; IQR: interquartile interval; CKD: Chronic Kidney Disease; HIV: human immunodeffiency virus; HBV: hepatitis B virus; HCV: hepatitis C virus; CMV: cytomegalovirus; cPRA: calculated panel reactive antibody; HLA mm: human leukocyte antigen mismatches; ECD: Expanded criteria donor; CNS: central nervous system; KDPI: Kidney Donor Profile Index.

episodes of acute rejection. Overall, there was no difference in the total number of treated acute rejection episodes (rATG-5, n = 30 vs. rATG-3, n = 28), yet the total number of antibody mediated and mixed acute rejection episodes in the rATG-3 was higher compared to rATG-5 group (11 vs. 4, Table 3), respectively. There was also no difference in the incidence or specific causes of graft loss and death.

Because 36 patients in the rATG group did not receive the intended total dose due to adverse events, a subgroup analysis revealed a higher incidence of first treated acute rejection (30.6% vs. 18.8%, p = 0.178) and first BCAR ≥ IA (19.4 vs. 4.7%, p = 0.033) comparing patients receiving < 5 mg/kg (n = 36) or ≥ 5 mg/kg (n = 64) total dose of rATG in the rATG-5 group.

The number of treated acute rejection episodes and graft losses from month 12 to month 36 were comparable but a higher number of deaths were observed in the rATG-5 compared to rATG-3 (S2 Table). There were no differences in cumulative survival-free of first BPAR (12 months: 90.0% vs. 86.4%, p = 0.477; 36 months: 74 vs. 80%, p = o.367, Fig 2A), patient survival

**Table 2. Immunosuppression.**

| Parameters | rATG-5 (n = 100) | rATG-3 (n = 110) | p value |
|---|---|---|---|
| rATG dose (mg/kg), median (IQR) | 5.30 (4.44; 6.16) | 3.01 (2.9; 3.11) | 0.000 |
| Immunosuppression, n (%) | | | |
| *Tacrolimus* | 99 (99) | 109 (99) | 0.946 |
| *Cyclosporine* | 1 (1) | 0 | 0.293 |
| *Mycophenolate* | 98 (98) | 98 (89) | 0.010 |
| *Azathioprine* | 0 | 3 (2) | 0.096 |
| *Everolimus* | 2 (2) | 9 (8.2) | 0.045 |
| Tacrolimus (ng/ml), mean±SD | | | |
| *Month 1* | 10.2±3.6 | 11.6±4.5 | 0.018 |
| *Month 3* | 8.6±3.0 | 9.7±4.9 | 0.117 |
| *Month 6* | 8.2±2.8 | 9.0±3.7 | 0.124 |
| *Month 12* | 7.8±2.8 | 9.1±3.5 | 0.008 |
| Mycophenolate (mg/day), mean±SD | | | |
| *Month 1* | 1349±230 | 1358±220 | 0.785 |
| *Month 3* | 1203±335 | 1231±325 | 0.604 |
| *Month 6* | 1228±318 | 1160±3511 | 0.306 |
| *Month 12* | 1221±313 | 1117±350 | 0.110 |
| Prednisone (mg/day), mean±SD | | | |
| *Month 1* | 17.4±6.4 | 12.4±5.4 | 0.000 |
| *Month 3* | 8.2±5.8 | 5.2±1.4 | 0.000 |
| *Month 6* | 6.1±3.6 | 5.2±0.9 | 0.039 |
| *Month 12* | 5.6±2.6 | 5.3±2.0 | 0.187 |

r-ATG: rabbit antithymocyte globulin; IQR: interquartile range; SD: standard deviation.

(12 months: 94.0% vs. 97.3%, p = 0.419; 36 months: 91.0% vs. 96.4%, p = 0.122, Fig 2B), graft survival (12 months: 88.0 vs. 89.1%, p = 0.843; 36 months: 76.0 vs. 80.0%, p = 0.539, Fig 2C), and death-censored graft survival (12 months: 93.0% vs. 91.8%, p = 0.944; 36 months: 84.0% vs. 83.6%, p = 0.948, Fig 2D) between the rATG-5 and rATG-3 groups, respectively.

## Safety assessments

Patients in the rATG-3 group tended to show a lower incidence of surgical complications compared to rATG-5, although there were no differences in the length of hospital stay for the transplant surgery or in the incidence and specific causes of early hospital readmission. The prevalence of high risk donor positive recipient negative CMV serostatus (D+/R-) combination was low and comparable between the two groups (Table 1). Using only preemptive therapy, there were no differences in the incidence of CMV events (48% vs. 40%, p = 0.19). None of the 11 patients receiving everolimus in both groups developed CMV infection, as well as the 3 patients who received azathioprine in the rATG-3 group. There was no difference in the incidence of CMV infection comparing patients receiving CNI/MPA in the rATG-5 (49%) and in the rATG-3 (45%). Importantly, most events were CMV infection and there were no episodes of CMV tissue invasive disease (Table 4).

## Delayed graft function, renal function and histology

There were no differences in the incidence and duration of DGF between the two groups. Patients in the rATG-5 group showed a significantly higher mean eGFR from month 1 to

**Table 3. Efficacy parameters at 12 months.**

| Parameters | rATG-5 (n = 100) | rATG-3 (n = 110) | p value |
|---|---|---|---|
| Efficacy failure, n (%)* | 18 (18) | 24 (21.8) | 0.522 |
| First BPAR $\geq$ IA, n (%) | 10 (10) | 15 (13.6) | 0.55 |
| First BPAR, n (%) | 20 (20) | 19 (17.3) | 0.616 |
| Severity *n (%)* | | | 0.305 |
| Borderline | 10 (43.5) | 4 (19.0) | |
| IA | 4 (17.4) | 6 (28.6) | |
| IB | 3 (13.0) | 1 (4.8) | |
| IIA | 2 (8.7) | 2 (9.5) | |
| ABMR | 1 (4.3) | 3 (14.3) | |
| Mixed | 0 (0) | 3 (14.3) | |
| Clinical acute rejection, n | 3 (13) | 1 (5.0) | |
| First tAR, n (%) | 23 (23) | 20 (18.2) | 0.485 |
| Time to first tRA, days, median (IQR) | 35 (13; 83) | 39 (10; 106) | 1.000 |
| Treatment, n (%) | | | 0.449 |
| Methylprednisolone | 20 (87.0) | 13 (61.9) | |
| rATG/methylprednisolone | 2 (8.6) | 4 (19.2) | |
| Plasmapheresis/Immunoglobulin | 1 (4.3) | 3 (14.3) | |
| Graft removal | 0 | 1 (4.8) | |
| Patients with recurrent AR, n (%) | 5 (21.7) | 6 (28.6) | 0.601 |
| Second rejection, n (%) | | | 0.239 |
| Borderline | 1 (20.0) | 2 (33.3) | |
| IA | 0 (0) | 0 (0) | |
| IB | 1 (20.o) | 0 (0) | |
| IIA | 0 (0) | 0 (0) | |
| ABMR | 2 (40.0) | 0 (0) | |
| Mixed | 1 (20.0) | 4 (66.7) | |
| Third rejection, n (%) | | | 0.466 |
| Borderline | 1 (25) | 0 (0) | |
| IA | 1 (25) | 0 (0) | |
| IB | 0 (0) | 1 (50.0) | |
| IIA | 0 (0) | 0 (0) | |
| ABMR | 0 (0) | 0 (0) | |
| Mixed | 0 (0) | 1 (50.o) | |
| Clinical acute rejection | 2 (50.0) | 0 (0) | |
| Graft loss, n (%) | 8 (8) | 10 (9) | 0.873 |
| FSFS recurrence | 1 (8.3) | 1 (7.1) | |
| Vascular thrombosis | 3 (37.5) | 2 (14.3) | |
| Primary Nonfunction | 1 (8.3) | 1 (7.1) | |
| Refractory acute rejection | 1 (8.3) | 2 (14.3) | |
| Pyelonephritis | 0 | 2 (14.3) | |
| Sepsis | 1 (8.3) | 1 (7.1) | |
| Withdrawal of immunosuppression | 1 (8.3) | 1 (7.1) | |
| Death, n (%) | 5 (5) | 4 (3.6) | 0.626 |
| Sepsis | 4 (80) | 2 (50) | |
| Acute myocardial infarction | 1 (20) | 1 (25) | |
| Traffic accident | 0 | 1 (25) | |

*(Continued)*

**Table 3.** (Continued)

| Parameters | rATG-5 (n = 100) | rATG-3 (n = 110) | p value |
|---|---|---|---|
| Loss to follow up, n (%) | 1 (1) | 5 (4.5) | 0.117 |

\* HR = 1.22, 95% CI 0.66–2.25.

r-ATG: rabbit antithymocyte globulin; BPAR: biopsy proven acute rejection; ABMR: antibody mediated rejection; IQR: interquartile range; tAR: treated acute rejection; AR: acute rejection; FSGS: focal and segmentar glomerulosclerosis; HR: hazard ratio; CI: confidence interval.

month 36 compared to patients in the rATG-3 group, with and without imputation, and mean urinary protein concentration was higher in the rATG-3 group at months 1, 24 and 36 (Table 5). The difference in eGFR was observed in patients without acute rejection, from month 1 (56.96±29.85 vs. 44.96±25.60 ml/min/1.73 m$^2$, P<0.001) to month 36 (53.25±28.22 vs. 44.58±24.00, p<0.001, respectively (S3 Table). There were no significant differences in mean eGFR of recipients of living donor allografts, and of deceased donors allografts with KDPI < 20% (Table 5, S2 Fig). Finally, there were no significant differences in renal function trajectories over the study period (Fig 3). The differences in eGFR at month 1 and month 12 were influenced by the unbalanced distribution of donor and recipient sex mismatches (S4 Table). Yet only the rATG-3 group (OR 2.93) and DGF (OR 2.46) were associated with inferior (< 48 ml/min/1.73 m$^2$) eGFR at 1 month (Table 6).

A higher proportion of patients in the rATG-3 group required at least one biopsy, either early or late after transplantation, and the number of biopsies per patient was also higher compared to patients in the rATG-5 group. Biopsies of patients in the rATG-3 group showed higher cg scores, early (0–3 months) and late (4–36 months) after transplantation. Nonetheless, there was no statistical difference in mean total Banff scores, either in biopsies performed within the first 3 months or afterwards (Table 7).

## Discussion

This sequential cohort experiment showed that the use of a single 3 mg/kg dose of rATG was associated with similar efficacy for the prevention of acute rejection, graft loss and death

### Cumulative incidence of efficacy failure

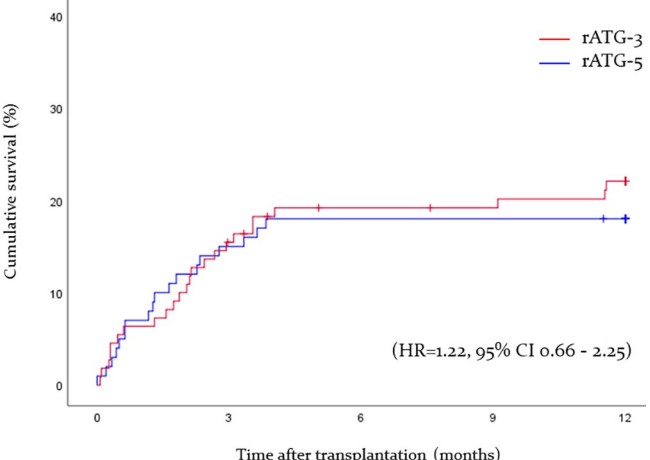

**Fig 1. Cumulative survival-free of efficacy failure (first biopsy proven acute rejection, graft loss, death), in patients undergoing retransplantation receiving 5 or 3 mg/kg rATG induction therapy.**

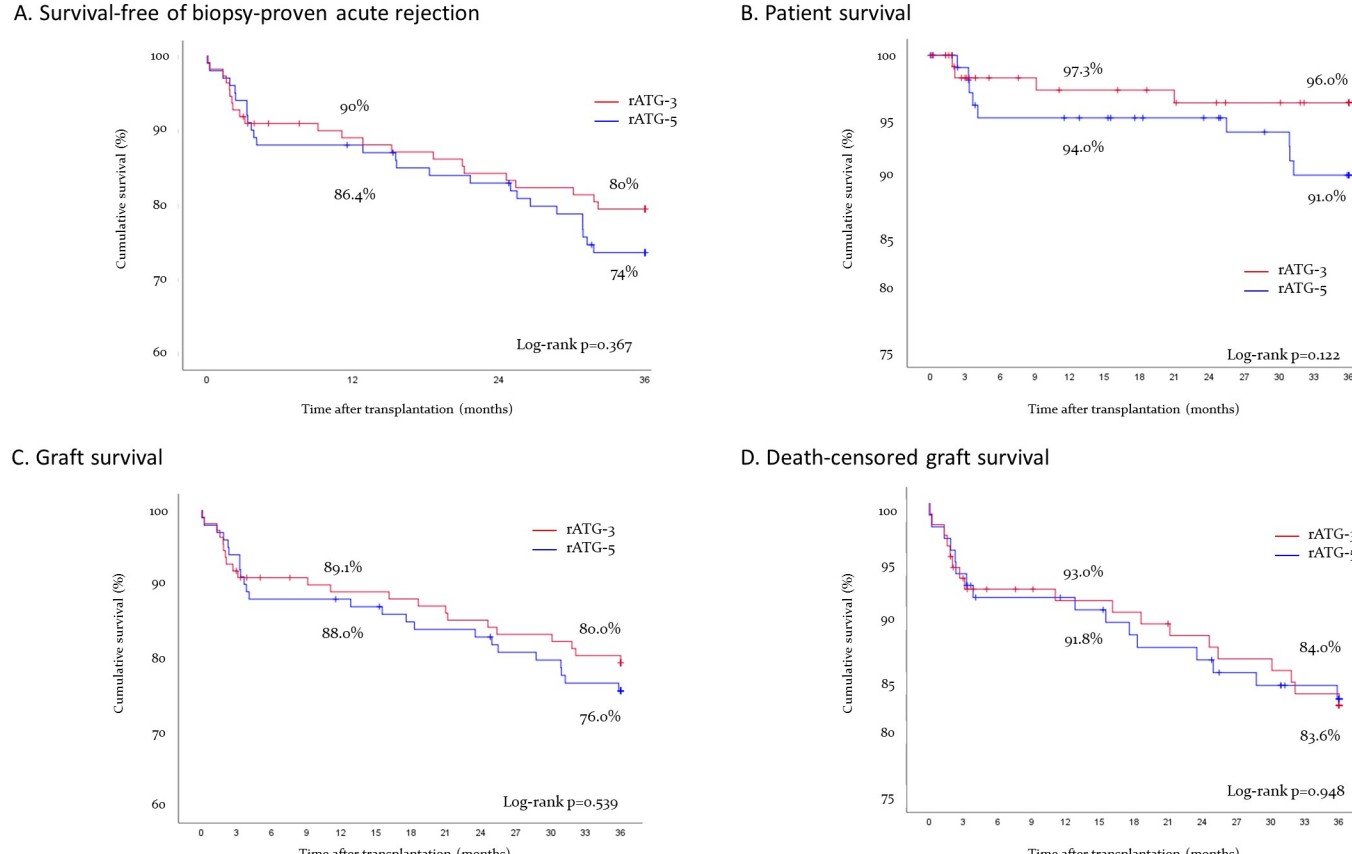

**Fig 2.** Cumulative survival-free of first biopsy proven acute rejection (A), patient survival (B), graft survival (C) and death-censored graft survival (D), in patients undergoing retransplantation receiving 5 or 3 mg/kg rATG induction therapy.

compared to the traditional 5 days course of 1 mg/kg rATG, over the first 3 years in recipients of repeat kidney transplants without preformed HLA A, B, and DR DSA. The observed 3-years graft, death-censored graft and patient survivals were comparable with that reported by the SRTR data [14]. These data are concordant with two other retrospective analysis in low-risk [15] and high-risk [16] kidney transplant recipients in our institution.

The demographic characteristics of the study population were comparable regarding key immunological risk factors. Furthermore, given the allocation policy and the large size of the local waiting list, 14% received a zero HLA A, B, DR mismatched and 73% zero HLA DR mismatched allografts, suggesting a further long-term benefit for this population [17, 18]. Recent studies have shown that the risk of antibody-mediated rejection and premature graft loss is associated with preformed DSA and not with the level of sensitization, measure by cPRA, as previously suggested [19]. Also, recent data suggest that, in the absence of detectable preformed DSA, reexposure to mismatched HLA antigens present in the first kidney transplant is not associated with de novo DSA development, rejection, or allograft loss [5]. Finally, the proportion of patients with previous allograft nephrectomy, which has been associated with an increased risk of development of anti-HLA antibodies [20], was comparable between the two groups.

In this cohort of patients we did not compare the pharmacodynamic effect of the two dosing rATG regimens using serial lymphocyte counts in the peripheral blood during the first year. A previous study showed comparable T, B and NK cell depletion up to one month in a

**Table 4. Surgical complications, early hospital readmission and CMV events.**

| Parameters | rATG-5 (n = 100) | rATG-3 (n = 110) | p value |
|---|---|---|---|
| Patients with surgical complication, n (%) | 25 (25) | 20 (18.2) | 0.229 |
| Time to first surgical complication (days), median (IQR) | 20 (5.5; 34.5) | 12.5 (0; 26.5) | 0.264 |
| Recurrent surgical complication, n (%) | 5 (20) | 4 (20) | 1.000 |
| Length of hospital stay for the transplant surgery, median (IQR) | 12.5 (4; 21) | 11 (6; 16) | 0.134 |
| Early hospital readmission, n (%) | 27 (27.8) | 31 (29.5) | 0.457 |
| Etiology, n (%) | | | 0.699 |
| Infection | 34 (61.8) | 39 (63.9) | |
| Acute rejection | 2 (3.6) | 5 (8.2) | |
| Surgical complication | 9 (16.4) | 8 (13.1) | |
| Cardiovascular | 6 (10.9) | 5 (8.2) | |
| Neoplasia | 1 (1.8) | 0 | |
| Toxicities | 3 (5.5) | 2 (3.3) | |
| Other | 0 | 2 (3,3) | |
| First CMV event, n (%) | 48 (48) | 44 (40) | 0.190 |
| Infection | 38 (78) | 33 (75) | |
| Disease | 11 (22) | 11 (25) | |
| Time to first CMV event, days, median (IQR) | 43 (30.7; 55.2) | 43,5 (26; 61) | 0,408 |
| Recurrent CMV event, n (%) | 9 (18.4) | 12 (27.3) | 0,308 |

r-ATG: rabbit antithymocyte globulin; IQR: interquartile range; CMV: cytomegalovirus.

cohort of patients receiving 3 or 6 mg/kg of rATG that persisted up to 12 months only in the 6 mg/kg dose group [21]. Importantly, only 64% of the patients in the rATG-5 receive the full 5 mg/kg course of rATG. This observation is frequent as shown in the seminal study by Brennan D et al., where only 68.6% of the patients received the intended five doses of rATG [22]. We also observed a similar pattern analyzing this strategy in a larger cohort of high-risk kidney transplant recipients [16]. The reasons to reduce the intended total dose are primarily safety issues, such as leukopenia, thrombocytopenia and surgical complications [22]. The trends towards higher incidence of acute rejection among patients receiving < 5 mg/kg in the rATG-5 group suggests that the inability to complete the intended 5 dose course, as a consequence of impeding toxicity, may increase the risk of acute rejection.

The incidence of first biopsy proven acute rejection was comparable to that observed in other cohorts [8]. This result is consistent with our previous study in high risk kidney transplant recipients, where a single 3 mg/kg rATG dose was associated with a decreased risk of CMV infection, without increasing the risk of acute rejection or compromising graft or patient survival compared to the traditional 6 mg/kg dose [16]. Yet, there were a higher number of antibody-mediated and mixed rejections in the rATG-3 group that might be associated, at least in part, with the presence of preformed anti-HLA C, DQ and DP antibodies [23].

Interestingly, the 2 mg/kg reduction in rATG dose was not associated with detectable safety benefits, including the number of surgical complications [24], early hospital readmissions and CMV infection. Nevertheless, a significant reduction in the incidence of CMV infection was observed in another high risk population using the same strategy in our center [16]. The overall incidence of CMV events was relatively high in both groups, but this is expected for patients receiving rATG without pharmacological prophylaxis [6]. Yet, despite the unique preemptive therapy strategy, more that 75% of the patients were treated for CMV infection and there were no cases of confirmed CMV invasive disease.

**Table 5. Incidence and duration of DGF, trajectories of renal function and proteinuria.**

| Parameters | rATG-5 (n = 100) | rATG-3 (n = 110) | p value |
|---|---|---|---|
| DGF, n (%) | 59 (59.6) | 64 (58.7) | 0.897 |
| *Duration of DGF, days, median (IQR)* | 4 (0; 10) | 2 (0; 8) | 0.188 |
| eGFR (ml/min/1.73 m$^2$, with imputation) | | | |
| *Month 1* | 54.7±28.8 | 44.1±25.3 | p = 0.005 |
| *Month 3* | 58.3±27.5 | 46.9±24.4 | p = 0.001 |
| *Month 6* | 57.7±27.4 | 47.5±29.4 | p = 0.05 |
| *Month 12* | 56.5±27.5 | 45.2±25.5 | p = 0.002 |
| *Month 24* | 51.4±27.9 | 42.8±24.7 | p = 0.019 |
| *Month 36* | 51.1±27.7 | 42.5±24.5 | p = 0.019 |
| eGFR (ml/min/1.73 m$^2$, without imputation) | | | |
| *Month 1* | | | |
| *All* | 55.4±27.9 | 44.1±25.3 | p = 0.003 |
| *Living donors* | 69.4±19.3 | 72.5±23.5 | p = 0.799 |
| *Deceased donors without DGF* | 60.0±27.16 | 49.3±26.2 | p = 0.157 |
| *Deceased donors KDPI 0–20%* | 54.8±31.1 | 58.6±26.4 | p = 0.548 |
| *Month 3* | 59.1±26.4 | 46.9±24.4 | p = 0.001 |
| *Month 6* | 59.9±25.6 | 47.3±24.4 | p = 0.001 |
| *Month 12* | | | |
| *All* | 58.5±25.8 | 44.6±25.0 | p = 0.000 |
| *Living donors* | 73.8±16.4 | 71.0±16.8 | p = 0.761 |
| *Deceased donors without DGF* | 58.5±24.6 | 44.4±24.3 | p = 0.022 |
| *Deceased donors KDPI 0–20%* | 66.1±27.9 | 52.4±19.6 | p = 0.127 |
| *Month 24* | 59.0±22.5 | 47.1±20.9 | p = 0.000 |
| *Month 36* | 58.9±21.4 | 49.1±19.5 | p = 0.003 |
| Proteinuria (g/mL) | | | |
| *Month 1* | 0.42±0.76 | 0.63±0.85 | p = 0.001 |
| *Month 3* | 0.40±0.76 | 0.40±0.51 | p = 0.087 |
| *Month 6* | 0.42±1.07 | 0.28±0.39 | p = 0.639 |
| *Month 12* | 0.50±1.72 | 0.40±0.83 | p = 0.306 |
| *Month 24* | 0.26±0.52 | 0.51±0.99 | p = 0.003 |
| *Month 36* | 0.38±0.81 | 0.70±2.40 | p = 0.008 |

DGF: delayed graft function; IQR: interquartile range; eGFR: estimate graft filtration rate; KDPI: Kidney Donor Profile Index.

The relative high but comparable incidence of DGF [25] is relevant in the context of our current allocation policy, because the concomitant presence of preformed DSA has been associated with higher incidence of subclinical mixed and antibody-mediated rejection phenotypes and inferior 5-years graft survival [26]. An intriguing observation was noticed in the rate of renal function recovery after retransplantation. There was an early, evident and persistent 10 ml/min lower eGFR in the rATG-3 group. There are several evidences suggesting that the unbalanced rejection rates between the two groups did not influence kidney function. First, the incidence of treated rejection was low in both groups (20% rATG-5 vs. 17.3% rATG-3). Second, the difference in kidney function was noticed as early as 1 month, when not all rejections had occurred, and persisted throughout the follow up. Thirdly, the trajectories of kidney function in patients without treated acute rejection showed the same pattern as the overall population. Subtle differences in demographic characteristics may be involved, independently of the development of delayed graft function and acute rejection [27], including the effect of

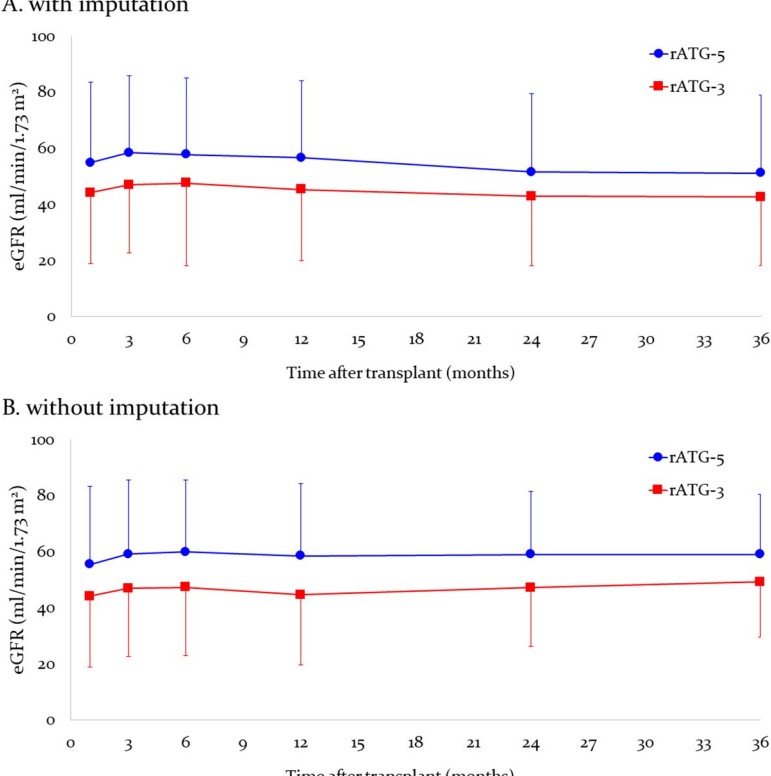

**Fig 3. Renal function trajectories (eGFR) over the first three years after kidney retransplantation, comparing patients receiving 5 and 3 mg/kg rATG induction therapy.** Data represents mean and standard deviation of eGFR with imputation (zero value for patients with graft loss, and last observation carried forward for patients who died or were loss to follow up).

donor-recipient sex mismatches on renal function and graft survival [28, 29]. It is also conceivable that the higher rATG dose showed higher efficacy to reduce the ischemia/reperfusion injury, promoting more complete recovery of graft function, primarily among those patients

**Table 6. Multivariable logistic regression analysis for 1 month eGFR < 48 ml/min/1.73 m$^2$ (n = 210).**

| Parameters | HR | 95% C.I. | | Sig. |
|---|---|---|---|---|
| | | Inferior | Superior | |
| KDPI, % | 1.012 | 0.998 | 1.026 | 0.083 |
| CIT, hours | 0.975 | 0.924 | 1.029 | 0.352 |
| SEX (M-M, reference) | | | | 0.611 |
| F-F | 1.051 | 0.429 | 2.576 | 0.913 |
| M-F | 1.102 | 0.464 | 2.619 | 0.263 |
| F-M | 0.263 | 0.614 | 0.262 | 0.826 |
| Time on dialysis, months | 1.005 | 0.999 | 1.011 | 0.132 |
| cPRA, % | 0.987 | 0.412 | 2.362 | 0.976 |
| Induction era (rATG-5, reference) | 3.471 | 1.829 | 6.589 | 0.000 |
| DGF (no, reference) | 2.549 | 1.300 | 4.997 | 0.006 |

eGFR: estimate graft filtration rate; HR: hazard ratio; CI: confidence interval; KDPI: Kidney Donor Profile Index; CIT: Cold ischemia time; M: Male; F: Female; cPRA: calculated panel reactive antibody; rATG: rabbit antithymocyte globulin; DGF: delayed graft function.

**Table 7. Summary of kidney transplant pathology of all biopsies up to 36 months.**

| Parameters | rATG-5 (n = 100) | rATG-3 (n = 110) | p value |
|---|---|---|---|
| Patients with biopsy, n (%) | 47 (47) | 78 (70.9) | 0.001 |
| Number of biopsies per patient, median (IQR) | 1 (1; 2) | 1 (1; 3) | 0.007 |
| Number of biopsies (0–3 months) | 45 | 110 | |
| Chronic Banff scores (0–3 months), mean±SD | | | |
| cg | 0.0 ± 0.0 | 0.1 ± 0.10 | 0.016 |
| ct | 0.21 ± 0.41 | 0.42 ± 0.65 | 0.112 |
| ci | 0.21 ± 0.41 | 0.46 ± 0.65 | 0.112 |
| cv | 0.39 ± 0.55 | 0.48 ± 0.72 | 0.131 |
| ah | 0.50 ± 0.65 | 0.41 ± 0.68 | 0.128 |
| cg+ct+ci+cv | 0.81 ± 1.10 | 1.33 ± 1.68 | 0.229 |
| Number of biopsies (4–36 months) | 41 | 77 | |
| Chronic Banff scores (4–36 months), mean±SD | | | |
| cg | 0.0 ± 0.0 | 0.04 ± 0.26 | 0.044 |
| ct | 1.17 ± 0.79 | 1.31. ±0.87 | 0.173 |
| ci | 1.17 ± 0.79 | 1.31. ± 0.87 | 0.173 |
| cv | 0.97 ± 0.97 | 0.97 ± 0.93 | 0.199 |
| ah | 0.79 ± 0.88 | 0.76 ± 0.84 | 0.176 |
| cg+ct+ci+cv | 3.22 ±2.15 | 3.61 ± 2.29 | 0.477 |

r-ATG: rabbit antithymocyte globulin; IQR: interquartile range; SD: standard deviation; cg: glomerular basement membrane double contours; ct: tubular atrophy; ci: interstitial fibrosis; cv: vascular fibrous intimal thickening; ah: arteriolar hyalinosis.

receiving high KDPI kidneys with long cold ischemia time [30]. Another possibility would be a higher incidence of subclinical rejections. Yet, as a consequence of incomplete recovery of kidney function, a higher number of early biopsies were performed in patients in the rATG-3, as demonstrated by the difference in GFR at month 1.

Because of the perceived higher risk for acute rejection among recipients of retransplants, small increases in creatinine triggered the indication of a biopsy to rule out rejection. Despite undergoing a higher number of kidney biopsies, there was no statistical difference in the number of treated acute rejections or in the acute Banff scores. Finally, the analysis of the chronic Banff scores suggests that the difference in kidney function is associated, at least in part, by the chronic cg scores observed in both early and late biopsies. Considering the stable eGFR trajectories over the 36 months, the early and persistent low-grade proteinuria, and the higher cg Banff scores in the rATG-3 group, we speculate that the initial difference in eGFR was derived from the interaction between subtle unfavorable donor characteristics—including age, lower proportion of male and consequent unbalanced donor/recipient match, donor acute kidney injury as measured by Δ creatinine—and longer cold ischemia time, and recovery from ischemia reperfusion injury.

There are numerous limitations in our study. There are no data regarding the outcomes of previous first and second transplants, including the dose of rATG, the incidence of DGF, level of renal function, survival [2], and reexposure to HLA mismatched antigens [5], all associated with clinical outcomes after retransplantation [2]. We also do not have data on HLA C, DP and DQ mismatches nor the presence of DSA against these antigens, which are becoming more relevant for both short and long-term graft survivals [31]. Further, we did not obtained serial measurements peripheral blood lymphocytes and DSA to ascertain whether the development of de novo DSA is influenced by the pharmacodynamics effect of the rATG dose [32,

33]. Currently, we are not only obtaining these data, but also performing a flow cytometry crossmatch in all sensitized patients, during the allocation process. We also do not have serial data on poliomavirus viremia, although poliomavirus nephropathy was captured by the for cause biopsies. While we only performed biopsies to investigate graft dysfunction, protocol biopsies would improve the ability to detecting subclinical inflammation [26].

In summary, this retrospective sequential cohort study was unable to detect differences in the incidence of efficacy failure and short-term surrogate safety assessments by reducing the dose of rATG from 5 to 3 mg/kg, in recipients of repeat kidney transplants without preformed HLA A, B, DR DSA. Considering the time course of the early and persistent difference in eGFR and the biopsy findings, the lower renal function in the rATG-3 group appears to derive from unfavorable donor characteristics during recovery from ischemia reperfusion injury rather than lack of efficacy for the prevention of clinical and subclinical rejection.

## Supporting information

**S1 Fig. Patient disposition.**
(TIF)

**S2 Fig. Mean month 1 eGFR stratified by KDPI scores, according to the rATG dose.**
(TIF)

**S1 Table. Distribution of rATG doses in the rATG-5 group.**
(DOCX)

**S2 Table. Acute rejection, graft loss and deaths from 12 to 36 months.**
(DOCX)

**S3 Table. Trajectories of kidney function stratified by the presence of treated acute rejection (tAR).**
(DOCX)

**S4 Table. Renal function stratified by donor and recipient gender combinations.**
(DOCX)

**S1 Appendix. Informed consent form for solid organs and tissues donation–deceased donor–over 18 years-old.**
(DOCX)

**S2 Appendix. Term of authorization for kidney donation from transplantation.**
(DOCX)

**S3 Appendix. Master file.**
(XLSX)

## Author Contributions

**Conceptualization:** Kamilla Linhares, Helio Tedesco-Silva.

**Data curation:** Kamilla Linhares, Henrique Proença.

**Formal analysis:** Kamilla Linhares, Julia Bernardi Taddeo, Klaus Nunes Ficher, Helio Tedesco-Silva.

**Investigation:** Kamilla Linhares.

**Methodology:** Kamilla Linhares, Marina Pontello Cristelli, Henrique Proença, Klaus Nunes Ficher, Renato de Marco, Maria Gerbase-DeLima, Jose Medina-Pestana, Helio Tedesco-Silva.

**Project administration:** Kamilla Linhares, Helio Tedesco-Silva.

**Supervision:** Kamilla Linhares, Helio Tedesco-Silva.

**Visualization:** Kamilla Linhares, Julia Bernardi Taddeo, Helio Tedesco-Silva.

**Writing – original draft:** Kamilla Linhares, Marina Pontello Cristelli, Helio Tedesco-Silva.

**Writing – review & editing:** Kamilla Linhares, Julia Bernardi Taddeo, Marina Pontello Cristelli, Henrique Proença, Klaus Nunes Ficher, Renato de Marco, Maria Gerbase-DeLima, Jose Medina-Pestana, Helio Tedesco-Silva.

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
