## [Decision Letter · Decision Letter 0]

8 Mar 2021

PONE-D-21-00777

The influence of the Antithymocyte Globulin dose on clinical outcomes of patients undergoing kidney retransplantation

PLOS ONE

Dear Dr. Tedesco-Silva,

Thank you for submitting your manuscript to PLOS ONE. After careful consideration, we feel that it has merit but does not fully meet PLOS ONE’s publication criteria as it currently stands. Therefore, we invite you to submit a revised version of the manuscript that addresses the points raised during the review process.

This manuscript is of interest to the renal transplant community. However, it is not acceptable for publication in its current form.

The manuscript requires more detailed methodology and results description as pointed out by both Reviewers, especially:

The power of the study to detect a difference in the primary endpoint should be stated, or are the author’s wanting to demonstrate non-inferiorityThe study population, baseline characteristics and outcome assessment is unclear as described in detail by Reviewer 3.It is unclear what dose the 5mg/kg group actually received because doses were increased or decreased on changes in lymphocyte count after each dose. The actual dose given should be reported.Please explain the details of loss to follow-up, e.g., reasons and causes. How was loss to follow up handled in analyses?The much higher biopsy rate in the 3 mg/kg group needs better explanation.

There are also other issues, which are described in detail by the Reviewers.

We look forward to receiving your revised manuscript.

Kind regards,

Justyna Gołębiewska

Academic Editor

PLOS ONE

Journal Requirements:

2. In your ethics statement in the manuscript and in the online submission form, please ensure that you have discussed whether all data/samples were fully anonymized before you accessed them and/or whether the IRB or ethics committee waived the requirement for informed consent.

If patients provided informed written consent to have data/samples from their medical records used in research, please include this information.

3. In the ethics statement in the manuscript and in the online submission form, please provide additional information about the patient records/samples used in your retrospective study, including:

a) the date range (month and year) during which patients' medical records/samples were accessed;

b) the source of the medical records/samples analyzed in this work (e.g. hospital, institution or medical center name).

4. We note that your study involved tissue/organ transplantation. Please provide the following information regarding tissue/organ donors for transplantation cases analyzed in your study.

1. Please provide the source(s) of the transplanted tissue/organs used in the study, including the institution name and a non-identifying description of the donor(s).

2. Please state in your response letter and ethics statement whether the transplant cases for this study involved any vulnerable populations; for example, tissue/organs from prisoners, subjects with reduced mental capacity due to illness or age, or minors.

- If a vulnerable population was used, please describe the population, justify the decision to use tissue/organ donations from this group, and clearly describe what measures were taken in the informed consent procedure to assure protection of the vulnerable group and avoid coercion.

- If a vulnerable population was not used, please state in your ethics statement, “None of the transplant donors was from a vulnerable population and all donors or next of kin provided written informed consent that was freely given.”

3. In the Methods, please provide detailed information about the procedure by which informed consent was obtained from organ/tissue donors or their next of kin. In addition, please provide a blank example of the form used to obtain consent from donors, and an English translation if the original is in a different language.

4. Please indicate whether the donors were previously registered as organ donors. If tissues/organs were obtained from deceased donors or cadavers, please provide details as to the donors’ cause(s) of death.

5. Please provide the participant recruitment dates and the period during which transplant procedures were done (as month and year).

6. Please discuss whether medical costs were covered or other cash payments were provided to the family of the donor. If so, please specify the value of this support (in local currency and equivalent to U.S. dollars).

6. Thank you for stating the following in the Competing Interests section:

'I have read the journal's policy and the authors of this manuscript have the following competing interests:

Helio Tedesco-Silva has received speaker’s fees and travel or accommodation expenses for development of educational presentations and scientific advice from Novartis, Pfizer, and Roche.

Jose Medina Pestana has received speaker’s fees and travel or accommodation expenses for development of educational presentations and scientific advice from Bristol-Myers Squibb, Novartis, Pfizer, and Roche.

Marina Pontello Cristelli has received speaker’s fees for development of educational presentations and travel or accommodation expenses from Novartis and Pfizer.

The other authors of this manuscript have no conflicts of interest to disclose.'

a. Please confirm that this does not alter your adherence to all PLOS ONE policies on sharing data and materials, by including the following statement: "This does not alter our adherence to  PLOS ONE policies on sharing data and materials.” (as detailed online in our guide for authors http://journals.plos.org/plosone/s/competing-interests).  If there are restrictions on sharing of data and/or materials, please state these.

Please note that we cannot proceed with consideration of your article until this information has been declared.

Reviewers' comments:

Reviewer's Responses to Questions

**Comments to the Author**

1. Is the manuscript technically sound, and do the data support the conclusions?

Reviewer #1: Yes

Reviewer #3: Partly

2. Has the statistical analysis been performed appropriately and rigorously? 

Reviewer #1: Yes

Reviewer #3: Yes

3. Have the authors made all data underlying the findings in their manuscript fully available?

Reviewer #1: Yes

Reviewer #3: Yes

4. Is the manuscript presented in an intelligible fashion and written in standard English?

Reviewer #1: Yes

Reviewer #3: Yes

5. Review Comments to the Author

Reviewer #1: This study examines the influence of the Antithymocyte Globulin dose on clinical outcomes of patients undergoing kidney retransplantation. The authors compared the efficacy and safety of rATG-5 and rATG-3. The outcomes of kidney transplantation were evaluated comprehensively. However, the research question seems not timely as the FDA has approved rATG as an induction drug with typical dose recommendations of 1.5 mg/kg in 2018. Studies have already compared efficady and safety between rATG-3 with rATG-1.5. Perhaps it isn't an urgent topic now to compare rATG-5 to rATG-3. Below are my minor comments about study design and statistical analysis:

1. Authors mentioned that this is a natural experiental study. However, time bias might be induced by following two comparison groups in different time periods.

2. Ethical approval for this study should be mentioned in the data source section.

3. Lack of statsitical significance for comparisons could be due to the rate events in two groups. Please calculate the statistical power for the major outcomes and discuss it in limitations.

4. Please explain the details of loss to follow-up, e.g., reasons and causes. How was loss to follow up handled in analyses?

5. Table 6 is only for 1 month eGFR < 48 ml/min/1.73 m2. The interaction of rATG and time can be examined in the model, then check the diffences of eGFR by rATG at each time point.

.

Reviewer #3: Overall: The study addresses an issue for which there is limited high quality data. It suffers from the usual limitations of retrospective studies. The power of the study to detect a difference in the primary endpoint should be stated, or are the author’s wanting to demonstrate non-inferiority – this should be clarified. There a number of clarifications/corrections needed -see details below. I think the study despite its limitations will be of some interest to readers, but I do not think the retrospective data will change practice. The much higher biopsy rate in the 3 mg/kg group needs better explanation, as does this higher rate of biopsy would like negate any economic benefit of the lower dosing.

Significance: There is little information about alternative dosing strategies for T cell depleting therapy. It is unlikely that a RCT will be conducted and registry data do not include dosing information -therefore single or multi-center retrospective series are likely to be the best available source of information.

Background Suggested Improvements:

As outlined in the introduction, retransplant patients are known to have worse outcomes than first transplant recipients and are considered to be at increased immunological risk. The authors may want to consider expanding on the reasons behind the increased immunological risk (intro or discussion) highlighting the potential roles of increased sensitisation and repeat class 2 HLA mismatches (e.g. Tinkham et al 2016). They may also consider providing data on the increasing incidence of retransplant as a proportion of all transplants undertaken (to support the relevance of this study) and also briefly summarise previous studies that have investigated induction regimens in retransplant recipients (e.g. Schold et al, 2015), and moreover outcomes with different ATG dosing regimens in transplantation in general (e.g singh et al, 2018). I feel having this information within the manuscript somewhere will give the reader greater context for the current study.

The current study takes advantage of a change in policy in a large single center and compares the standard of care (SOC 5X 1mg/kg dosing used in the period Jan 2010- Jun 2014) versus a novel single 3 mg / kg dose (used in the period June 2014 – Oct 16) in retransplant recipients of a living or deceased donor transplant

The key finding was no difference in the composite efficacy outcome of biopsy proven acute rejection, death or graft loss and no difference in safety indicators including 30 day re-hospitalization, CMV infection/disease.

There were 100 / 110 patients in each group with 3 year post transplant outcomes reported.

Although this is a retrospective study – it is not clear what the type II error is in this relatively small study cohort

In addition to the usual limitations of retrospective studies there are a number of clarifications required to assess this study:

1. The study population is unclear: a) The number transplanted – number excluded does not add up to the number included in the study. 2) The study protocol states that study patients in the 1 mg/kg group all had cPRA >50% - but there are clearly patients in this group with cPRA <50%. The method of assessment for DSA should be provided.

2. Baseline characteristics – it is not clear what initial and final Cr and delta Cr mean in Table 1. How was KDPI determined ? – against what reference population? I am not aware that Brazil uses a KDPI calculator. Were there no DCD donor kidneys used?

3. It is unclear what dose the 5mg/kg group actually received because doses were increased or decreased on changes in lymphocyte count after each dose. The actual dose given should be reported – it may be that a significant % of those in the 5 mg/kg group actually received 3 mg/kg or even less. As lymphopenia is an important outcome, it would be important to know the incidence of this in both groups.

4. Outcome assessment – the protocol states all biopsies for rejection were determined using the Banff 2019 criteria – this cannot be the case as most of the patients were included before 2019. What criteria were used to determine/classify AR. Was there retrospective review of all AR biopsies?

5. Outcome assessment – what was the test platform used for CMV antigenemia determination and what were the test parameters. If these changed over time it may be better to report those requiring ganciclovir treatment.

6. Outcome assessment -as there was an imbalance in Everolimus maintenance between groups – and EVR is known to be protective against CMV – was the incidence of CMV different between groups among patients treated with CNI/MPA

7. Outcome assessment – the far higher bx rate in the 3mg/kg cohort requires explanation

8. There were more humoral rejections in the 3 mg/kg group – the authors should convince us that the lower overall level of kidney function in the 3mg/kg group is not due to this.

9. The authors should confirm if any/what other changes in practice occurred over this timeframe. We note, for example, that the change in practice was based on avoiding transplantation in patients with preformed antibody to HLA A, B and DR antigens – does this mean that in the cohort prior to the change patients were being potentially transplanted in the presence of such antibodies (albeit CDC XM neg)? Was flow XM undertaken at any stage during the study? Were any other changes undertaken during the period to improve HLA matching? The authors state that in the first period all patients with PRA >50% received ATG whereas PRA wasn’t taken in to consideration in the second period. This should be clarified as table 1 demonstrates the some patients in the first period have PRA under 50%. Together, this suggests the cohorts may have been of unequal immunological risk? Please address.

6. PLOS authors have the option to publish the peer review history of their article (what does this mean?). If published, this will include your full peer review and any attached files.

Reviewer #1: No

Reviewer #3: No

---

## [Author Response · Author response to Decision Letter 0]

25 Mar 2021

Dear Justyna Gołębiewska

Academic Editor

PLOS ONE 

Thank you the comments and suggestion provide to our manuscript 

PONE-D-21-00777: The influence of the Antithymocyte Globulin dose on clinical outcomes of patients undergoing kidney retransplantation. Please find below a point-by-point reply to each of the editors’ and reviewers’ comments.

Editor

This manuscript is of interest to the renal transplant community. However, it is not acceptable for publication in its current form.

The manuscript requires more detailed methodology and results description as pointed out by both Reviewers, especially:

1. The power of the study to detect a difference in the primary endpoint should be stated, or are the author’s wanting to demonstrate non-inferiority.

YES! The aim was to demonstrated non-inferiority. This was added to the methods section (see below).

2. The study population, baseline characteristics and outcome assessment is unclear as described in detail by Reviewer 3.

We clarified all issues raised by the reviewers regarding study population, baseline characteristics and outcome assessments (see below).

3. It is unclear what dose the 5 mg/kg group actually received because doses were increased or decreased on changes in lymphocyte count after each dose. The actual dose given should be reported. 

YES. We now provide the actual doses given (see below). 

4. Please explain the details of loss to follow-up, e.g., reasons and causes. How was loss to follow up handled in analyses?

We explained the details of loss to follow-up and how was it was handled in the analyses.

5. The much higher biopsy rate in the 3 mg/kg group needs better explanation.

Detailed explanation regarding the higher biopsy rate in the 3 mg/kg group is provided below.

Journal Requirements:

We formatted our manuscript according to the PLOS ONE style requirements. 

2. In your ethics statement in the manuscript and in the online submission form, please ensure that you have discussed whether all data/samples were fully anonymized before you accessed them and/or whether the IRB or ethics committee waived the requirement for informed consent.

The research project was approved by the local Research Ethics Committee of the Federal University of São Paulo (CAAE 02285018.2.5505). The IRB waived the requirement to obtain informed consent because of the retrospective nature of the record review, the lack of interference with the rights and welfare, and the lack of risk to the participants. All data were retrieved from the electronic medical records and were fully anonymized before analysis. Medical records of the Hospital do Rim (Fundação Oswaldo Ramos) were accessed between 01/06/2017 and 01/10/2020.

If patients provided informed written consent to have data/samples from their medical records used in research, please include this information.

 Not applicable

3. In the ethics statement in the manuscript and in the online submission form, please provide additional information about the patient records/samples used in your retrospective study, including:

a) the date range (month and year) during which patients' medical records/samples were accessed;

Medical records were accessed between 01/06/2017 and 01/10/2020. 

b) the source of the medical records/samples analyzed in this work (e.g. hospital, institution or medical center name).

 Medical records of the Hospital do Rim (fundação Oswaldo Ramos) were accessed between 01/06/2017 and 01/10/2020. 

4. We note that your study involved tissue/organ transplantation. Please provide the following information regarding tissue/organ donors for transplantation cases analyzed in your study.

1. Please provide the source(s) of the transplanted tissue/organs used in the study, including the institution name and a non-identifying description of the donor(s).

Kidney transplant were performed at the Hospital do Rim with organs recovered from deceased donors under the National and São Paulo State secretary allocation policy, and from living donors, according to the National legislation. 

2. Please state in your response letter and ethics statement whether the transplant cases for this study involved any vulnerable populations; for example, tissue/organs from prisoners, subjects with reduced mental capacity due to illness or age, or minors.

- If a vulnerable population was used, please describe the population, justify the decision to use tissue/organ donations from this group, and clearly describe what measures were taken in the informed consent procedure to assure protection of the vulnerable group and avoid coercion.

- If a vulnerable population was not used, please state in your ethics statement, “None of the transplant donors was from a vulnerable population and all donors or next of kin provided written informed consent that was freely given.”

3. In the Methods, please provide detailed information about the procedure by which informed consent was obtained from organ/tissue donors or their next of kin. In addition, please provide a blank example of the form used to obtain consent from donors, and an English translation if the original is in a different language.

Informed consent was obtained from all donors or their next of kin, prior to organ recovery. Translated informed consent forms for deceased and living donors are now provided as supplemental material. 

4. Please indicate whether the donors were previously registered as organ donors. If tissues/organs were obtained from deceased donors or cadavers, please provide details as to the donors’ cause(s) of death.

The national regulation does not mandate registration of donors. The deceased donors causes of death are provided in Table 1. 

5. Please provide the participant recruitment dates and the period during which transplant procedures were done (as month and year).

Record data from kidney transplants performed between 01Jan2010 to 09Oct2016 were screened for this analysis.

6. Please discuss whether medical costs were covered or other cash payments were provided to the family of the donor. If so, please specify the value of this support (in local currency and equivalent to U.S. dollars).

According to Brazilian legislation, no medical costs or other cash payments are allowed to the family of the donor.

There are no restrictions

The anonymized data set necessary to replicate our study findings is provided as S3-Appendix-Master file. 

6. Thank you for stating the following in the Competing Interests section:

'I have read the journal's policy and the authors of this manuscript have the following competing interests:

Helio Tedesco-Silva has received speaker’s fees and travel or accommodation expenses for development of educational presentations and scientific advice from Novartis, Pfizer, and Roche.

Jose Medina Pestana has received speaker’s fees and travel or accommodation expenses for development of educational presentations and scientific advice from Bristol-Myers Squibb, Novartis, Pfizer, and Roche.

Marina Pontello Cristelli has received speaker’s fees for development of educational presentations and travel or accommodation expenses from Novartis and Pfizer.

The other authors of this manuscript have no conflicts of interest to disclose.'

a. Please confirm that this does not alter your adherence to all PLOS ONE policies on sharing data and materials, by including the following statement: "This does not alter our adherence to PLOS ONE policies on sharing data and materials.” 

(as detailed online in our guide for authors http://journals.plos.org/plosone/s/competing-interests). If there are restrictions on sharing of data and/or materials, please state these.

Please note that we cannot proceed with consideration of your article until this information has been declared.

There is no new information to update in the Competing Interests statement.

Reviewers' comments:

Reviewer's Responses to Questions

Comments to the Author

1. Is the manuscript technically sound, and do the data support the conclusions?

Reviewer #1: Yes

Reviewer #3: Partly

2. Has the statistical analysis been performed appropriately and rigorously?

Reviewer #1: Yes

Reviewer #3: Yes

3. Have the authors made all data underlying the findings in their manuscript fully available?

Reviewer #1: Yes

Reviewer #3: Yes

4. Is the manuscript presented in an intelligible fashion and written in standard English?

Reviewer #1: Yes

Reviewer #3: Yes

5. Review Comments to the Author

Reviewer #1: This study examines the influence of the Antithymocyte Globulin dose on clinical outcomes of patients undergoing kidney retransplantation. The authors compared the efficacy and safety of rATG-5 and rATG-3. The outcomes of kidney transplantation were evaluated comprehensively. However, the research question seems not timely as the FDA has approved rATG as an induction drug with typical dose recommendations of 1.5 mg/kg in 2018. Studies have already compared efficacy and safety between rATG-3 with rATG-1.5. Perhaps it isn't an urgent topic now to compare rATG-5 to rATG-3. Below are my minor comments about study design and statistical analysis:

While the FDA approved the use of rATG for induction therapy, the dosing regimen is still open for debate, depending on the immunological risk and the risk of developing delayed graft function. Our main goal was to demonstrate that the use of a single 3 mg/kg dose of rATG would provide comparable efficacy to the traditional 5 mg/kg dose, in high immunological risk patients undergoing retransplantation but without pre-formed HLA A, B, DR donor specific antibodies. 

1. Authors mentioned that this is a natural experiental study. However, time bias might be induced by following two comparison groups in different time periods.

YES! We changed the study design as a sequential cohort study.

2. Ethical approval for this study should be mentioned in the data source section.

Ethical approval for this study is reported in the Ethics section.

3. Lack of statistical significance for comparisons could be due to the rate events in two groups. Please calculate the statistical power for the major outcomes and discuss it in limitations.

YES! The sample size lack sufficient statistical power considering the significance event rate events in the two groups. In fact, this was a non-inferiority trial and detailed of the sample size and power were now added. The following paragraph was added to the statistical analysis section.

“Considering the non-inferiority margin of 15% for the experimental rATG-3 group compared to the standard rATG-5 group for the primary endpoint of efficacy failure at 12 months, a significance level of 5%, a statistical power of 80%, and the efficacy failure of 25% in the control arm, 208 patients are required to be 80% sure that the upper limit of a one-sided 95% confidence interval. We used Cox proportional hazard model to compare hazard functions using treatment group as factor. Non-inferiority was determined if upper confidence limit of HR was less than the non-inferiority margin then non-inferiority using Wald’s confidence interval.”

4. Please explain the details of loss to follow-up, e.g., reasons and causes. How was loss to follow up handled in analyses?

The only cause of loss to follow up was referral to another transplant center. For survival analysis, patients with loss to follow up were censored. For comparison of the trajectories of renal function, last observation carried forward analysis was used. 

5. Table 6 is only for 1 month eGFR < 48 ml/min/1.73 m2. The interaction of rATG and time can be examined in the model, then check the differences of eGFR by rATG at each time point.

Apologies for our lack of clarity. Table 6 includes all patients in both groups (n=210) and we performed a multivariable logistic regression to identify the risk factors associated with reaching 1 month with an eGFR < 48 ml/min/1.73 m2. We believe this is the most appropriate statistical approach. In addition, as shown in Figure 3, the trajectories of renal function are parallel from month. Therefore, differences of eGFR at each time point persisted throughout the first year. 

Reviewer #3: Overall: The study addresses an issue for which there is limited high quality data. It suffers from the usual limitations of retrospective studies. The power of the study to detect a difference in the primary endpoint should be stated, or are the author’s wanting to demonstrate non-inferiority – this should be clarified. There a number of clarifications/corrections needed -see details below. I think the study despite its limitations will be of some interest to readers, but I do not think the retrospective data will change practice. The much higher biopsy rate in the 3 mg/kg group needs better explanation, as does this higher rate of biopsy would like negate any economic benefit of the lower dosing.

YES! The sample size lack sufficient statistical power considering the event rate in the two groups. In fact, this was a non-inferiority trial and detailed of the sample size and power were now added.

We provided several evidences suggesting that the higher rate of biopsies in the 3 mg/kg group was associated with donor characteristics (see below), although. The perceived benefits of the lower dose are (1) improved dosing schedule: only one dose within the first 24 hours after graft revascularization; (2) improved safety monitoring: during the infusion that patients are in an intensive monitoring unit; (3) earlier hospital discharge before day 5; (4) shorter duration of lymphopenia (unfortunately this was not investigated in this study – see reference 18); (5) increased safety, not seen in this analysis, but observed in another high-risk population using the same strategy (reference 19). 

Significance: There is little information about alternative dosing strategies for T cell depleting therapy. It is unlikely that a RCT will be conducted and registry data do not include dosing information -therefore single or multi-center retrospective series are likely to be the best available source of information.

We fully agree with the reviewer. We decided to use universal 3 mg/kg rATG induction therapy, except for recipients of HLA identical living donor kidney transplants, after an initial data obtained from a prospective randomized trial (reference 12). We then allowed sufficient time to gather minimum data suggesting that this strategy is effective and safe in low-risk (reference 15) and high-risk (reference 16) kidney transplant recipients. We are now describing the initial experience with retransplants. Altogether these data suggests that this strategy is feasible but, of course, cannot be immediately extrapolated to other populations without careful monitoring. 

Background Suggested Improvements:

As outlined in the introduction, retransplant patients are known to have worse outcomes than first transplant recipients and are considered to be at increased immunological risk. The authors may want to consider expanding on the reasons behind the increased immunological risk (intro or discussion) highlighting the potential roles of increased sensitisation and repeat class 2 HLA mismatches (e.g. Tinkham et al 2016). They may also consider providing data on the increasing incidence of retransplant as a proportion of all transplants undertaken (to support the relevance of this study) and also briefly summarise previous studies that have investigated induction regimens in retransplant recipients (e.g. Schold et al, 2015), and moreover outcomes with different ATG dosing regimens in transplantation in general (e.g singh et al, 2018). I feel having this information within the manuscript somewhere will give the reader greater context for the current study.

We agree with the reviewer. We included these data in the introduction and added the suggested references.

The current study takes advantage of a change in policy in a large single center and compares the standard of care (SOC 5X 1mg/kg dosing used in the period Jan 2010- Jun 2014) versus a novel single 3 mg / kg dose (used in the period June 2014 – Oct 16) in retransplant recipients of a living or deceased donor transplant

The key finding was no difference in the composite efficacy outcome of biopsy proven acute rejection, death or graft loss and no difference in safety indicators including 30 day re-hospitalization, CMV infection/disease.

There were 100 / 110 patients in each group with 3 year post transplant outcomes reported.

Although this is a retrospective study – it is not clear what the type II error is in this relatively small study cohort.

YES! The sample size lack sufficient statistical power considering the event rate in the two groups. In fact, this was a non-inferiority trial and detailed of the sample size and power were now added. This limitation was highlighted in the discussion. 

In addition to the usual limitations of retrospective studies there are a number of clarifications required to assess this study:

1. The study population is unclear: a) The number transplanted – number excluded does not add up to the number included in the study. 

YES! The numbers were not accurate. Please see below the correction. We also included a Supplemental Figure 1 with the disposition of the study population.

“There were 4030 kidney transplants between January 1, 2010 and June 16, 2014, of which 203 (5%) were retransplants, while among 2098 procedures between June 17, 2014 and October 9, 2016, 139 (6.6%) were retransplants. We excluded 103 retransplants in the first period and 29 in the second period who did not receive induction therapy with rATG (S4-Figure 1).”

2) The study protocol states that study patients in the 1 mg/kg group all had cPRA >50% - but there are clearly patients in this group with cPRA <50%. 

YES! This was not clear. cPRA was in fact calculated retrospectively for the comparison between the two groups in this analysis. Our institutional protocol used 1 mg/kg of rATG for patients with PRA class I or class II > 50%. 

While all recipients with PRA>50% received up to 5 doses of 1 mg/kg of rATG, patients with PRA< 50% could receive the same induction strategy based on other risk factors such as HLA compatibility, cold ischemia time, priority criterion, and donor type. The proportion of patients with PRA class I or class II >50% was 65% (n=65) in the rATG-5 group and 56% (n=62) in the rATG-3 group. This information was added to the methods and result sections. 

The method of assessment for DSA should be provided.

To identify anti-HLA antibodies in serum, FlowPRA screening test is performed, followed by single antigen bead assay to identify the specificity of anti-HLA antibodies, using the Luminex platform.

2. Baseline characteristics – it is not clear what initial and final Cr and delta Cr mean in Table 1. 

Initial creatinine value is the first value obtained at the time of hospital admission of the potential donor. Final creatinine is the last creatinine value before organ recovery. Delta creatinine is an arbitrary measure of acute kidney injury. This clarification was added as a footnote in Table 1.

How was KDPI determined? – against what reference population? I am not aware that Brazil uses a KDPI calculator. 

KDPI was calculated using the UNOS formula developed using registry data from the North American kidney transplant population. KDPI has not been validated in Brazil as none of the other indexes developed in North America (for example, the older “extended criteria donor” definition). We used KPDI as a more granular measure of kidney donor “quality” for simple comparison between the two groups 

Were there no DCD donor kidneys used?

In Brazil, the use of organ from DCD donors is forbidden by law. 

3. It is unclear what dose the 5mg/kg group actually received because doses were increased or decreased on changes in lymphocyte count after each dose. The actual dose given should be reported – it may be that a significant % of those in the 5 mg/kg group actually received 3 mg/kg or even less. As lymphopenia is an important outcome, it would be important to know the incidence of this in both groups.

In the rATG 5 mg/kg group, 2 patients received 1, 3 patients received 2, 1 patient received 3, 11 patients received 4, 80 patients received 5, and 3 patients received 6 doses of rATG. 

In the rATG 5 mg only 4 patients received less than 3 mg/kg, 32 received between 3 and 5 mg/kg, 32 received between 5 and 6 mg/kg, and 32 more than 6 mg/kg S5- Table 1). 

4. Outcome assessment – the protocol states all biopsies for rejection were determined using the Banff 2019 criteria – this cannot be the case as most of the patients were included before 2019. What criteria were used to determine/classify AR. Was there retrospective review of all AR biopsies?

All biopsies with rejection were retrospectively reclassified using the Banff 2019 criteria. 

5. Outcome assessment – what was the test platform used for CMV antigenemia determination and what were the test parameters. If these changed over time it may be better to report those requiring ganciclovir treatment.

During the study period, the CMV antigenemia test was performed in a single laboratory using the CMV Brite Turbo kit, according to the manufacturer's recommendations (IQ® Products, Groningen, Netherlands). The data provided (Table 4) refers to patients treated with ganciclovir meeting the protocol defined criteria for CMV infection or disease.

6. Outcome assessment -as there was an imbalance in Everolimus maintenance between groups – and EVR is known to be protective against CMV – was the incidence of CMV different between groups among patients treated with CNI/MPA.

None of the patients receiving EVR in both groups developed CMV infection, as well as the 3 patients who received azathioprine in the rATG-3 group. 

Therefore, there was no difference incidence of CMV infection comparing patients receiving CNI/MPA in the rATG-5 (49%, 48/98) and in the raTG-3 (45%, 44/98). 

7. Outcome assessment – the far higher bx rate in the 3mg/kg cohort requires explanation.

The higher rate of kidney biopsies in the 3 mg/kg dose group is a consequence of early and incomplete recovery of kidney function after transplantation, as demonstrated by the difference in GFR at month 1. Because of the perceived higher risk for acute rejection among recipients of retransplants, small increases in creatinine triggered the indication of a biopsy to rule out rejection. Despite undergoing a higher number of kidney biopsies, there was no statistical difference in the number of treated acute rejections. Finally, the analysis of the chronic Banff scores suggests that the difference in kidney function is associated, at least in part, by the chronic cg scores observed in both early and late biopsies.

8. There were more humoral rejections in the 3 mg/kg group – the authors should convince us that the lower overall level of kidney function in the 3mg/kg group is not due to this.

There are several evidences suggesting that the unbalanced rejection rates between the two groups did not influence kidney function. First, the incidence of treated rejection was low in both groups (20% rATG-5 vs. 17.3% rATg-3). Second, the difference in kidney function was noticed as early as 1 month (Figure 3), when not all rejection had occurred, and persisted throughout the follow up. Thirdly, the trajectories of kidney function in patients without treated acute rejection showed the same pattern as the overall population (S7- Table 3). 

9. The authors should confirm if any/what other changes in practice occurred over this timeframe. 

There were no further changes in the protocols or in clinical practice over the time frame of this study.

We note, for example, that the change in practice was based on avoiding transplantation in patients with preformed antibody to HLA A, B and DR antigens – does this mean that in the cohort prior to the change patients were being potentially transplanted in the presence of such antibodies (albeit CDC XM neg)? 

In both cohorts of patients (rATG-5 and rATG-3) patients with preformed antibody to HLA A, B and DR antigens were noy eligible for kidney transplantation.

Was flow XM undertaken at any stage during the study? 

No, only recently we started using flow XM.

Were any other changes undertaken during the period to improve HLA matching? 

NO.

The authors state that in the first period all patients with PRA >50% received ATG whereas PRA wasn’t taken into consideration in the second period. This should be clarified as table 1 demonstrates some patients in the first period have PRA under 50%. Together, this suggests the cohorts may have been of unequal immunological risk. Please address.

YES! This was not clear. While all recipients with PRA>50% received up to 5 doses of 1 mg/kg of rATG, patients with PRA< 50% could receive the same induction strategy based on other risk factors such as HLA compatibility, cold ischemia time, priority criterion, and donor type. The proportion of patients with PRA class I or class II >50% was 65% (n=65) in the rATG-5 group and 56% (n=62) in the rATG-3 group. This information was added to the methods and result sections.

---

## [Decision Letter · Decision Letter 1]

6 Apr 2021

PONE-D-21-00777R1

The influence of the Antithymocyte Globulin dose on clinical outcomes of patients undergoing kidney retransplantation

PLOS ONE

Dear Dr. Tedesco-Silva,

Thank you for submitting your manuscript to PLOS ONE. After careful consideration, we feel that it has merit but does not fully meet PLOS ONE’s publication criteria as it currently stands. Therefore, we invite you to submit a revised version of the manuscript that addresses the points raised during the review process.

ACADEMIC EDITOR:

Thank you for all the new information provided and for all explanations included. However, there are still issues that require clarification.

“In the rATG-5 group only 4 patients received less than 3 mg/kg, 32 received between 3 and 5 mg/kg” – 36 out of 100 patients in rATG-5 group did not receive 5mg/kg. This is a source of a substantial bias and has not even been acknowledged in the discussion. This is a retrospective analysis, not a prospective randomized clinical trial, the patients should be divided into groups according to the actual dose they received and all results recalculated. Another way would be to exclude this 36 patients from the analysis, but as I understand the study would be then underpowered to even show non-inferiority.

Please define the following: „low HLA compatibility, long cold ischemia time, priority criterion, and donor type”

“Altogether, the stable eGFR trajectories over the 36 months, the early and persistent low-grade proteinuria, and the higher cg Banff scores in the rATG-3 group suggest that the initial difference in eGFR was derived from unfavorable donor characteristics and recovery from ischemia reperfusion injury.” – but according to data in Table 1 donor characteristics did not differ significantly between both groups, please clarify.

We look forward to receiving your revised manuscript.

Kind regards,

Justyna Gołębiewska

Academic Editor

PLOS ONE

Reviewers' comments:

Reviewer's Responses to Questions

**Comments to the Author**

1. If the authors have adequately addressed your comments raised in a previous round of review and you feel that this manuscript is now acceptable for publication, you may indicate that here to bypass the “Comments to the Author” section, enter your conflict of interest statement in the “Confidential to Editor” section, and submit your "Accept" recommendation.

Reviewer #1: All comments have been addressed

2. Is the manuscript technically sound, and do the data support the conclusions?

Reviewer #1: Yes

3. Has the statistical analysis been performed appropriately and rigorously? 

Reviewer #1: Yes

4. Have the authors made all data underlying the findings in their manuscript fully available?

Reviewer #1: Yes

5. Is the manuscript presented in an intelligible fashion and written in standard English?

Reviewer #1: Yes

6. Review Comments to the Author

Reviewer #1: Tnanks for addressing all reviewer comments. Your responses are appropriate. The revised manuscript is acceptable.

7. PLOS authors have the option to publish the peer review history of their article (what does this mean?). If published, this will include your full peer review and any attached files.

Reviewer #1: No

---

## [Author Response · Author response to Decision Letter 1]

8 Apr 2021

Dear Justyna Gołębiewska

Academic Editor

PLOS ONE

Thanks for the new insights provide to our manuscript. Please find below a point-by-point clarification of the pending issues 

ACADEMIC EDITOR:

Thank you for all the new information provided and for all explanations included. However, there are still issues that require clarification.

“In the rATG-5 group only 4 patients received less than 3 mg/kg, 32 received between 3 and 5 mg/kg” – 36 out of 100 patients in rATG-5 group did not receive 5mg/kg. This is a source of a substantial bias and has not even been acknowledged in the discussion. This is a retrospective analysis, not a prospective randomized clinical trial, the patients should be divided into groups according to the actual dose they received and all results recalculated. Another way would be to exclude these 36 patients from the analysis, but as I understand the study would be then underpowered to even show non-inferiority.

AUTHOR: YES! The editor is quite correct and this has to be highlighted in our manuscript. This could be a source of a substantial bias and has not even been acknowledged in the discussion. Below we discuss this possible bias, added more data and analysis, and acknowledge it in the discussion section.

In the seminal manuscript by Brennan D et al, which we use as reference and adapted to implement our previous standard of care induction therapy “rabbit antithymocyte globulin was initiated before reperfusion in 87.9% of all patients assigned to receive it, and 68.8% received the intended five doses.” [Brennan DC, Daller JA, Lake KD, Cibrik D, Del Castillo D; Thymoglobulin Induction Study Group. Rabbit antithymocyte globulin versus basiliximab in renal transplantation. N Engl J Med. 2006 Nov 9;355(19):1967-77. doi: 10.1056/NEJMoa060068. PMID: 17093248.] 

In fact, in all studies addressing alternative doses of r-ATG, the control group was the current “standard of care”. Yet, the actual doses are always lower than that described in the methods section in each manuscript, as all analyses were performed as “intention to treat”. 

We added the following paragraph to the methods section:

“All comparisons were made using the intention to treat population, defined as patients receiving at least of dose of rATG in both groups.”

The reasons to reduce the intended total dose are primarily safety issues, such as leukopenia, thrombocytopenia and surgical complications. The same issue was observed when we analyzed the same strategy in high-risk kidney transplant recipients (reference 16). The debate is even expanded as the pharmacodynamic measure (lymphopenia) does not correlate with dose and is not routinely used. 

Overall, it is difficult to determine whether the bias in favor or against the rATG-5 group. The inability to administer the 5 doses as scheduled could be interpreted as “treatment failure” because the limiting toxicity may increase the risk of acute rejection. The safety of the single 3 mg/kg dose approach is therefore an advantage compared to the 5 doses of 1 mg/kg regimen. 

We reviewed most of these manuscripts to determine a lower r-ATG dose that could provide a balanced efficacy/safety profile and ultimately tested this 3 mg/kg r-ATG single dose in a prospective trial (reference 12) before adopting it as our new “standard of care”. 

ACADEMIC EDITOR: The patients should be divided into groups according to the actual dose they received and all results recalculated. 

AUTHOR: We added the following paragraph to the result section:

Because 36 patients in the rATG group did not receive the intended total dose due to adverse events, a subgroup analysis revealed a higher incidence of first treated acute rejection (30.6% vs. 18.8%, p=0.178) and first BCAR ≥ IA (19.4 vs. 4.7%, p=0.033) comparing patients receiving < 5 mg/kg (n=36) or ≥ 5 mg/kg (n=64) total dose of rATG in the rATG-5 group.

We added the following paragraph to the discussion section:

Importantly, only 64% of the patients in the rATG-5 receive the full 5 mg/kg course of rATG. This observation is frequent as shown in the seminal study by Brennan D et al, where only 68.6% of the patients received the intended five doses of rATG (22). We also observed a similar pattern analyzing this strategy in a larger cohort of high-risk kidney transplant recipients (16). The reasons to reduce the intended total dose are primarily safety issues, such as leukopenia, thrombocytopenia and surgical complications (22). The trends towards higher incidence of acute rejection among patients receiving < 5 mg/kg in the rATG-5 group suggests that the inability to complete the intended 5 dose course, as a consequence of impeding toxicity, may increase the risk of acute rejection. 

ACADEMIC EDITOR: Please define the following: „low HLA compatibility, long cold ischemia time, priority criterion, and donor type”

AUTHOR: Low HLA compatibility was defined as more than 3 HLA mismatches.

Long cold ischemia time is defined as cold ischemia time higher than 24 hours

Priority criterion is the priority status attributed to a patient with imminent lack of access for peritoneal hemodialysis during the allocation process

Donor types are living, and standard or expanded criteria deceased donors

This information was added to the method section.

ACADEMIC EDITOR: “Altogether, the stable eGFR trajectories over the 36 months, the early and persistent low-grade proteinuria, and the higher cg Banff scores in the rATG-3 group suggest that the initial difference in eGFR was derived from unfavorable donor characteristics and recovery from ischemia reperfusion injury.” – but according to data in Table 1 donor characteristics did not differ significantly between both groups, please clarify.

AUTHOR: YES! The editor is correct. In fact, this paragraph is in our discussion as a speculation for the unexpected findings in kidney function. We rule out the role of acute rejection (considering that we were diligent in excluding this diagnosis by performing a higher number of biopsies in this group of patients). We were left with this interaction, “unfavorable donor characteristics and recovery from ischemia reperfusion injury”. We called “unfavorable donor characteristics” a combination of older age (42 vs. 45.5 years, p-0.421), lower male donors (65% vs. 49.1%, p=0.02) and consequent unbalanced donor/recipient match (p=0.043), donor acute kidney injury as measured by � creatinine (0.2 vs. 0.5 mg/dl, p=0.019), and longer cold ischemia time (22 vs. 24 hours, p=0.06). We stressed in the discussion that this is speculative.

---

## [Editor Report · Decision Letter 2]

26 Apr 2021

The influence of the Antithymocyte Globulin dose on clinical outcomes of patients undergoing kidney retransplantation

PONE-D-21-00777R2

Dear Dr. Tedesco-Silva,

We’re pleased to inform you that your manuscript has been judged scientifically suitable for publication and will be formally accepted for publication once it meets all outstanding technical requirements.

Kind regards,

Justyna Gołębiewska

Academic Editor

PLOS ONE
---

## [Editor Report · Acceptance letter]

3 May 2021

PONE-D-21-00777R2 

The influence of the Antithymocyte Globulin dose on clinical outcomes of patients undergoing kidney retransplantation. 

Dear Dr. Tedesco-Silva:

I'm pleased to inform you that your manuscript has been deemed suitable for publication in PLOS ONE. Congratulations! Your manuscript is now with our production department. 

Kind regards, 

on behalf of

Dr. Justyna Gołębiewska 

Academic Editor

PLOS ONE